# Maintenance Therapy for Pancreatic Cancer, a New Approach Based on the Synergy between the Novel Agent GP-2250 (Misetionamide) and Gemcitabine

**DOI:** 10.3390/cancers16142612

**Published:** 2024-07-22

**Authors:** Marie Buchholz, Britta Majchrzak-Stiller, Ilka Peters, Stephan Hahn, Lea Skrzypczyk, Lena Beule, Waldemar Uhl, Chris Braumann, Johanna Strotmann, Philipp Höhn

**Affiliations:** 1Department of General and Visceral Surgery, Division of Molecular and Clinical Research, St. Josef-Hospital, Ruhr-University Bochum, 44791 Bochum, Germany; britta.majchrzak-stiller@rub.de (B.M.-S.); ilka.peters@klinikum-bochum.de (I.P.); lea.skrzypczyk@klinikum-bochum.de (L.S.); lena.beule@rub.de (L.B.); waldemar.uhl@klinikum-bochum.de (W.U.); chris.braumann@evk-herne.de (C.B.); johanna.strotmann@ruhr-uni-bochum.de (J.S.); philipp.hoehn@klinikum-bochum.de (P.H.); 2Department of Molecular Gastrointestinal Oncology, Ruhr-University Bochum, 44780 Bochum, Germany; stephan.hahn@rub.de; 3Department of General, Visceral and Vascular Surgery, Evangelische Kliniken Gelsenkirchen, Akademisches Lehrkrankenhaus der Universität Duisburg-Essen, 45878 Gelsenkirchen, Germany

**Keywords:** maintenance, GP-2250, misetionamide, chemotherapy, pancreatic cancer, CD133

## Abstract

**Simple Summary:**

Pancreatic ductal adenocarcinoma (PDAC) is an aggressive cancer type with rising incidences worldwide and poor survival. GP-2250 is an emerging agent showing a high antineoplastic capacity. Currently, GP-2250 is under phase I clinical trial for PDAC. This study was the first to evaluate the antineoplastic effects of GP-2250 on pancreatic adenocarcinoma in combination with Gemcitabine in a PDAC patient-derived mouse xenograft (PDX) setting. This combination showed highly synergistic effects in a primary cell culture model, as well as in a first-line and a maintenance setting using PDX. Changes in the CD133 content of treated spheroid cultures provide first indicators for this synergism. These findings demonstrate the vast potential of this highly promising combination in the maintenance therapy of PDAC patients.

**Abstract:**

The novel Oxathiazinane derivative GP-2250 (Misetionamide) displays antineoplastic activity in vitro and in vivo, as previously shown in pancreatic cancer cells and in patient-derived mouse xenografts (PDX). Currently, GP 2250 is under phase I clinical trial in pancreatic ductal adenocarcinoma (PDAC). GP-2250 in combination with Gemcitabine displays a high synergistic capacity in various primary and established pancreatic cancer cell lines. Additionally, in the eight PDX models tested, the drug combination was superior in reducing tumor volume with an aggregate tumor regression (ATR) of 74% compared to Gemcitabine alone (ATR: 10%). Similarly, in a PDX maintenance setting following two weeks of treatment with nab-Paclitaxel plus Gemcitabine, the combination of GP-2250 plus Gemcitabine resulted in outstanding tumor control (ATR: 79%) compared to treatment with Gemcitabine alone (ATR: 19%). Furthermore, GP-2250 reduced the ratio of tumor-initiating CD133^+^ markers on the surface of PDAC cells in spheroid cultures, indicating a possible mechanism for the synergistic effect of both substances. Considering the high tolerability of GP 2250, these results may open up a new approach to maintenance therapy with GP-2250/Gemcitabine combination following nab-Paclitaxel plus Gemcitabine as first-line treatment.

## 1. Introduction

Pancreatic cancer is one of the most lethal gastrointestinal cancers and is usually diagnosed at an advanced stage when curative therapy is often not possible. It is the fourth most common cause of cancer-related death in Germany [1,2]. Pancreatic ductal adenocarcinoma (PDAC) is the most common type of pancreatic cancer worldwide. Due to a lack of early symptoms and diagnostic markers, PDAC is often diagnosed at a late stage. Additionally, only a few therapeutics were shown to be effective, and the rapid development of drug resistance combined with the high toxicity of these drugs is the main reason for the low 5-year survival rate of 9% [3,4,5,6].

Surgical resection is the only potentially curative therapy for pancreatic cancer [7,8]. Additional adjuvant chemotherapy according to the FOLFIRINOX protocol (5-Fluorouracil, Leucovorin, Irinotecan, Oxaliplatin), with Gemcitabine or 5-Fluorouracil (5-FU) is recommended [9], expanding the 5-year survival rate by 15–20%. However, only 15–20% of the tumors are resectable at the time of diagnosis [10]. In cases of non-resectable tumors, chemotherapy is used with palliative intention or to achieve secondary resectability in order to prolong survival and increase quality of life, respectively. For patients with a good performance status, FOLFIRINOX or nab-Paclitaxel/Gemcitabine are administered as first-line treatment, although FOLFIRINOX has a low safety profile with various severe side effects such as peripheral neuropathy [9,10,11]. For patients with lower overall conditions, Gemcitabine as monotherapy or in combination with Erlotinib or nab-Paclitaxel in a reduced dosage is used [9,10,12].

However, continued uninterrupted full-dose chemotherapy often leads to cumulative toxicity without additional efficacy [13,14]. Therefore, different maintenance strategies have been explored for advanced PDAC after sustained tumor control has been achieved [15,16]. One approach is the reduction in the number of chemotherapeutic agents used. Indeed, in clinical studies, neither the progression-free survival (PFS) rate nor the median overall survival (OS) was significantly improved by these maintenance strategies [17,18]. The other approach is to introduce another chemotherapeutic agent with a different mechanism of action, often mutation-targeted therapies, also known as switch maintenance [19,20,21,22,23,24]. Among these, the poly(adenosine diphosphate-ribose) polymerase (PARP) inhibitor Olaparib showed the most promising results in maintenance following platinum-based chemotherapy in metastasized PDAC patients with BRCA 1 or 2 germline mutation, increasing the mean progression-free survival from 3.8 to 7.4 months [19]. In Western countries, only 4.5 to 8% of unselected PDAC patients display BRCA 1 or 2 germline variants though [19,25], severely limiting the number of patients benefitting from targeted therapy. Hence, the development and research of new therapeutic opportunities, especially regarding emerging substances, are essential to improve patient survival and quality of life.

The novel substance GP-2250, known as Misotionamide, is an Oxthiazinane derivate (1.4.5-oxathiazan-dioxide-4.4), which is characterized by a cyclic six-membered ring structure containing a sulfur, an oxygen, and a nitrogen heteroatom (Figure 1). Recently, it has been demonstrated to possess antiproliferative, antineoplastic, and migration-inhibiting effects on several tumor entities such as PDAC, pancreatic neuroendocrine carcinoma, mesothelioma, colon and Merkel cell carcinoma, as well as mammarian cancer cells in vitro [26,27,28,29,30]. The substance is currently undergoing an open-label phase 1/2 clinical trial to evaluate the safety and tolerability of GP-2250 on patients with advanced unresectable or metastatic pancreatic adenocarcinoma (NCT03854110) [31]. Its mechanism of action is largely based on the impairment of the cancer cell energy metabolism by inhibiting HK2 (Hexokinase 2) and GAPDH (Glyceraldehyde 3-phosphate dehydrogenase) resulting in ATP reduction. Thereby, AMP-Kinase is activated leading to a downregulation of mTOR (mammalian Target of Rapamycin), thus reducing cell growth and proliferation. Furthermore, lower ATP levels induce reactive oxygen species (ROS) production, resulting in apoptosis via mitochondrial damage. GP-2250 also inhibits NFκB (Nuclear factor kappa light-chain enhancer of activated B cells) and thereby supports apoptosis induction via downregulation of Bcl-2 (B-cell lymphoma 2) and inhibition of cell proliferation by downregulating Cyclin D1 [26,27,28,29,30].

In vivo, GP-2250 shows a reduction in tumor growth in xenografts of established pancreatic cancer cell lines, as well as in patient-derived xenograft (PDX) models of pancreatic adenocarcinoma, accompanied by rarely occurring side effects [26,29]. Evaluation of the maximal tolerable dose revealed acute toxicity at 2000 mg/kg*BW and chronic toxicity at concentrations higher than 1000 mg/kg*BW in nude mice. No changes in body weight or vital function were observed at lower concentrations [26,29]. Crucially, a synergy between GP-2250 and Gemcitabine was discovered in the first study of pancreatic neuroendocrine carcinoma. GP-2250 in combination with Gemcitabine showed partial remission of the neuroendocrine tumor volume according to response evaluation criteria in solid tumors (RECIST) in an in vivo mouse model of PDX [29]. Nevertheless, the underlying mechanism of this synergy between Gemcitabine and GP-2250 remains to be determined. A clinically relevant factor, associated with poor therapeutic outcome and tumor recurrence after Gemcitabine treatment, is the expression of the surface marker CD133, also known as prominin-1 [32,33]. In 2007, Hermann et al. found that a CD133^+^ subgroup of PDAC cells was associated with a high ability for tumor formation and higher resistance towards chemotherapeutics like Gemcitabine [34]. Cells overexpressing CD133 were also found to have a higher glucose uptake and increased glycolysis, along with a higher NfκB expression [35,36]. Inhibition of glycolysis increased the sensitivity towards Gemcitabine of otherwise resistant CD133^+^ cells [35]. The impact of GP-2250 on CD133^+^ cells was therefore explored as a potential target for sensitizing tumors to Gemcitabine.

The present study aims to investigate the potential benefit of the combination of GP-2250 and Gemcitabine for PDAC patients. Firstly, it was examined whether the drug combination of GP-2250 and Gemcitabine displayed synergistic cytotoxicity in PDAC cell lines. Furthermore, the efficiency of this synergistic drug combination was tested in a PDX mouse model. Additionally, the drug combination was examined in a subsequent PDX model in a maintenance setting after a two-week treatment with nab-Paclitaxel plus Gemcitabine to analyze the effectiveness of tumor control. As GP-2250 targets the glucose metabolism and inhibits NFκB, we finally sought to examine its impact on CD133^+^ PDAC cells in spheroid culture to obtain a first indication of the mechanism of synergy between those two substances.

## 2. Materials and Methods

### 2.1. Cell Lines and Culture Conditions

Four different human pancreatic cancer cell lines were used for the in vitro experiments. The established cell line was PancTuI (ATCC—LGC Standards GmbH, Wesel, Germany). PDAC primary cell lines Bo73, Bo80, and Bo103 were isolated and established from PDX tissue, as described by Hermann et al. [34]. PancTuI and Bo80 cells were cultured in Dulbecco’s Modified Eagle Medium (DMEM) with 5% FCS, 100 U/mL Penicillin, 100 µg/mL Streptomycin, and 2 mM L-Glutamine (See Table 1). Bo73 and Bo103 were cultured in modified DMEM/F12 Medium [37]. The details of the ingredients are listed in Table 1. Cells were passaged for fewer than 6 months after establishment or receipt from the cell banks. Authentication was analyzed via STR analysis.

The cells were cultivated at 37 °C in a water vapor-saturated atmosphere at 5% CO_2_ (*v*/*v*) and were regularly screened for mycoplasma contamination.

### 2.2. Spheroid Cultivation

Spheroid cultivation was performed to increase the ratio of CD133^+^ cells using a free-floating method according to Miranda-Lorenzo et al. [38]. For the spheroid formation, 400,000 cells per well were seeded on ultra-low adhesion 6-well plates in 2 mL spheroid medium, where 1 mL of spheroid medium (DMEM/F12 (Pan Biotech, Aidenbach, Germany) supplemented with 100 U/mL Penicillin/Streptomycin, 1× MEM Non-essential Amino Acid (NEAA; Pan Biotech P08-32100), B27 (1:50, gibco, 17504044), 100 µg/mL bFGF (Thermo Fischer, Waltham, MA, USA, 13256-029) was added every two days (Table 1). The cultivation time until the formation of spheroids occurred varied depending on the cell line (Bo80 7–9 days, PancTuI 5–7 days). Spheroids were passaged and separated from single cells and loose cells aggregated with a 40 µm cell strainer and transferred back onto low-adhesion plates for cultivation, treatment, and further experiments.

### 2.3. Tissue

Thirteen different human pancreatic cancer tissues were used for the in vivo experiments. As every tumor has a unique mutation profile, 13 different PDX models were used in total for the in vivo studies (See Table 2). The selection for the two different treatment methods was made randomly and was not based on certain criteria.

### 2.4. Reagents

GP-2250 ultrapure powder (kindly provided by Geistlich Pharma AG, Wohlhusen, Switzerland) was dissolved in double-distilled water (ddH_2_O), sterile filtered, set to a physiological pH, and freshly prepared once weekly. Gemcitabine was purchased from the company Hexal (Novartis) (Holzkirchen, Germany), further diluted in physiological saline solution, and freshly prepared once weekly.

### 2.5. MTT Cytotoxicity Assay

Cells were seeded to a density of 3.5 × 104 cells/well in 96-well plates and incubated for 24 h to obtain a subconfluent monolayer. The cells were treated with increasing concentrations of GP-2250 and Gemcitabine for 48 h to determine the dose–response antineoplastic effect. In addition, the combinations of both agents were analyzed. Two hours prior to measurement, 10 µL of yellow MTT (3-(4.5-Dimethylthiazol-2-yl)-2.5-diphenyltetrazoliumbromide, Sigma Aldrich, St. Louis, MI, USA) reagent (5 mg/mL) was added into each well. Viable cells convert yellow MTT into violet formazan crystals. The test media was discarded, and 100 µL DMSO (Dimethylsulfoxide, Carl Roth, Karlsruhe, Germany) was applied. After an incubation time of 5–10 min, cell viability was analyzed using a microplate absorbance reader by measuring the absorption at 550 nm and 720 nm as the reference wavelength (Asys UVM 340, Biochrom, Berlin, Germany). The amount of violet formazan produced was directly proportional to the number of viable cells [39]. The assay was performed in 4–6 independent experiments with consecutive passages.

### 2.6. FACS Analysis

GP-2250 and Gemcitabine were evaluated for their effects on populations in spheroids of PancTuI and Bo80 incubated either with 1000 µM GP-2250 or with 34 mM Gemcitabine for 48 h compared to untreated cells serving as negative controls. To dissociate spheroids, they were centrifuged (4 min, 230 rpm) and rinsed with PBS before resuspension in Accumax cell dissociation solution (PAN Biotech GmbH, Aidenbach, Germany) after discarding the supernatant. Spheroids were incubated at 37 °C and pipetted several times every 10–15 min until dissociation occurred (25 to 60 min depending on the cell line). Separated cells were centrifuged and resuspended in staining buffer (PBS + 5% FCS) and incubated for 30 min without light with CD133/1 (AC133)-PE or Mouse IgG1k-PE (MACS Miltenyi Biotec, Bergisch Gladbach, Germany) (1:20) as a control for unspecific binding. Unstained cells were used as negative control to determine baseline cell population. Afterward, the cells were washed 2 times with 500 µL staining buffer to remove unbound antibodies and reduce background before flow cytometric analysis was performed. The laser excited 488 nm and the emission was measured at 520 nm (FACSCalibur, BD Biosciences, Franklin Lakes, NJ, USA). Cell detritus and fragments were distinguished by a forward and side scatter and excluded from the measurement. Data were analyzed using the Cell Quest ProTM software Version 5.1.

### 2.7. Animal Studies

Six-week-old female NMRI Foxn1nu/Foxn1nu mice (Janvier, Le Genest-Saint-Isle, France) were acclimated to a 12 h light cycle-controlled environment for at least one week before initiation of the study. The animals were provided standard laboratory food and water ad libitum. Mice were anesthetized by isoflurane inhalation. Tumor tissue fragments were implanted subcutaneously in the flank region [40,41]. After implantation, the recipient mice were monitored for general health status, body weight (BW), and the presence of subcutaneous tumors. Tumor volume was determined by measuring tumor diameters (measurement of 2 perpendicular axes of the tumor) twice weekly using a caliper and calculated as follows:V = (π/6) × (ab^2^),(a = larger axis, b = smaller axis).

Following randomization, the systemic influence of GP-2250, Gemcitabine, and their combination on tumor growth after intraperitoneal (i.p.) administration was investigated. Group 1: control treated with vehicle; Group 2: Gemcitabine 50 mg/kg*BW twice weekly; Group 3: GP-2250 500 mg/kg*BW three times per week; Group 4: combination therapy of GP-2250 500 mg/kg*BW three times per week and Gemcitabine 50 mg/kg*BW twice weekly on alternating days. Treatment was initiated when the tumor volume reached 250 mm^3^. Tumor volume was measured twice weekly. The experiment was terminated either after an application period of 9 weeks or when the tumor reached a volume of 1000 mm^3^. Therapy response was evaluated according to Response Evaluation Criteria In Solid Tumors (RECIST) [42].

### 2.8. Maintenance Study

Mice were randomized into three groups. Group 1: control treated with vehicle; Group 2: Gemcitabine 50 mg/kg*BW twice weekly; Group 3: combination therapy of GP-2250 500 mg/kg*BW three times per week and Gemcitabine 50 mg/kg*BW twice weekly on alternating days. This regime was initiated after previous treatment of all animals for two weeks with Gemcitabine 100 mg/kg*BW twice weekly and nab-Paclitaxel 30 mg/kg*BW twice weekly on alternating days, and the tumor volume was measured twice weekly. The experiment was terminated either after an application period of 8–10 weeks or when the tumor reached a volume of 1000 mm^3^. The therapy response was evaluated according to RECIST [42].

### 2.9. Immunhistochemival Validation of Therapy Response

H and E staining and regression grade analysis was performed of representative tumors (73, 69, 103, 122 first-line therapy). Slides were stained according to an established hematoxylin and eosin protocol (Mayer’s hemalum solution, WALDECK, Ref: 2E-010 Lot: 31151, Eosin 1% Waldeck 1B-425) [43,44]. All slides were analyzed by two pathologists who were blinded to all information regarding the tumors’ treatment [45,46]. The pathologic tumor regression grading was executed according to two established graduating systems: Le Scodan and College of American Pathologists (CAP). Both of the systems focus on the relation of therapy-induced changes of viable tumor mass to fibrosis.

Regression grading according to Le Scodan: Grade 1 (minor response): large distribution of vital tumor cells, less than 50% SDCC, no complete necrosis; Grade 2 (moderate response): large distribution of vital tumor cells, more than 50% SDCC, sporadically small necrosis; Grade 3 (partial response): few vital tumor cells, 80% of which are SDCC, extensive complete necrosis; Grade 4 (complete response): no vital tumor cells remain.

Regression grading according to CAP: Grade 0 (complete response): no vital tumor cells remain; Grade 1 (near complete response): few viable tumor cells (<5% viable tumor cells); Grade 2 (partial response): large areas of residual cancer with apparent tumor regression, predominance of fibrosis over tumor cells; Grade 3 (poor or no response): extensive residual cancer and large infiltrates of viable cancer cells, predominance of tumor cells over fibrosis.

### 2.10. Statistics and Calculations

The results of the MTT assay (percentage of viable cells) and characteristics of mice (body weight, tumor volume) are expressed as mean ± standard deviation (SD). Comparison between experimental groups with normal distribution were performed using one-way ANOVA followed by Tukey’s post hoc test, and pairwise tests were performed using *t*-tests. Fisher’s exact test was used for categorical data, if appropriate. Statistical significance was set at *p* ≤ 0.05, and is indicated in the figures as follows: *** *p* ≤ 0.001, ** *p* ≤ 0.01, * *p* ≤ 0.05, n.s. *p* > 0.05 (not significant). Calculations were performed using Graph Pad Prism 5.1.0 and 9.1.0.

The synergistic effects were analyzed by computing the combination index (CI) by Chou–Talalay using CalcuSyn, version 2.11 (Biosoft, Cambridge, UK). A CI < 0.95 was classified as a synergistic effect (+), CI = 0.95–1.1 as an additive effect (±), and CI > 1.1 as an antagonistic effect (−). In the resulting isobolograms of the software, the x- and y-axes reflect the doses of the individual components. The isobole is the line/curve between points of the same effect; therefore, a straight line means that the two substances have an additive effect. In the case of synergism, the overall effect of the two compounds is bigger than expected from the summation of their separate effects, resulting in a concave curve. Oppositely, antagonistic interaction displays, that the overall effect is less than expected from the summation of the separate effects and a convex curve is obtained. The mean effective dose was computed using CalcuSyn software version 2.0. This analysis was performed as described by Chou et al. [47,48].

### 2.11. Ethical Considerations

The local ethical committee approved the collection of sample tissue from patients with PDAC, as well as the implantation and expansion of cancer tissue in xenograft mouse models. Written informed consent was obtained from all patients according to the local ethics guidelines. This study was conducted in accordance with the Declaration of Helsinki. All procedures were performed according to a protocol approved by the Ethics Committee of Ruhr University Bochum (permission no. 2392 3. amendment). All animal experiments were performed in accordance with the guidelines of the local Animal Use and Care Committee (Permission No. 81-02.04.2018.A169).

## 3. Results

### 3.1. Synergy of GP-2250 and Gemcitabine in Pancreatic Cancer Cell Lines

In the first step, the impact of GP-2250 and Gemcitabine as single agents on cell viability was compared to their combination using the MTT viability assay in three primary cell lines (Bo103, Bo80, and Bo73) and the established cell line PancTul. When the three primary cell lines were treated with Gemcitabine alone (100 and 1000 µM), no significant loss of cell viability was observed (Figure 2A–C). In contrast, GP-2250 reduced the viability of all three cell lines. In Bo103, viability was reduced to 64.87 ± 6.31% with 500 µM and 20.46 ± 5.17% with 1000 µM GP-2250. In Bo80, viability was reduced to 85.37 ± 5.63% with 200 µM GP-2250 and to 5.74 ± 0.54% with 500 µM GP-2250. In Bo73 cells, cell viability was reduced to 48.28 ± 5.97% with 650 µM GP-2250.

The IC50 of GP-2250 after 48 h was determined using a linear regression model with five different concentrations of GP-2250. In cell line Bo103, IC50 was reached at 670 µM, Bo80 at 257 µM, and Bo73 at 646 µM. The primary cells were not sensitive enough to Gemcitabine to determine the IC50 after 48 h with a reasonable amount of substance. If possible, we used a dosage of medium-range cytotoxicity for GP-2250 with a cell viability between 30 and 60%.

However, when inactive doses of Gemcitabine were combined with GP-2250, a remarkable synergy became apparent in all cell lines tested. In Bo103, combining GP-2250 (500 µM) with Gemcitabine (1000 µM) reduced cell viability to 32.87 ± 5.72% (CI = 0.624), as compared to 64.87% with GP-2250 as a single agent (Figure 2A). A combination index (CI) of 0.624 indicated a synergistic effect. Likewise, synergy was achieved in Bo103 by combining 100 µM Gemcitabine with 1000 µM GP-2250, with a CI of 0.634. In Bo80, GP-2250 (200 µM) plus Gemcitabine (100 µM) reduced cell viability to 24.72 ± 5.84% (CI = 0.382). A similar result (27.73 ± 6.66%) was achieved with 200 µM GP-2250 and 1000 µM Gemcitabine (CI = 0.410), whereas GP-2250 as a single agent at 200 µM reduced viability to only 85.37 ± 5.63% (Figure 2B). A synergistic effect was also found for Bo73, where Gemcitabine (100 µM) plus GP-2250 (600 µM) reduced cell viability to 57.87 ± 7.67% (CI = 0.903), which contrasts with the reduction by GP-2250 as a single agent at the same concentration to 85.35 ± 7.37% (Figure 2C).

A synergistic effect was also observed for the established cell line PancTuI, which was considerably more sensitive to Gemcitabine as opposed to the primary cell lines. The IC50 value of GP-2250 after 48 h was 285 µM. The IC50 value of Gemcitabine was 115 µM after 48 h. Figure 2D shows a significant effect for all analyzed concentrations of both substances. The combination of 150 µM GP-2250 and either 0.1 µM Gemcitabine or 1 µM Gemcitabine reduced cell viability to 49.11 ± 6.41% (CI = 0.712) and 36.94 ± 7.11% (CI = 0.616) respectively, displaying a synergistic effect. The combinations of 0.1 µM Gemcitabine or 1 µM Gemcitabine with 200 µM GP 2250 further reduced cell viability to 24.51 ± 6.87% (CI 0.693) and 32.26 ± 8.25% (CI 0.774).

### 3.2. PDX In Vivo Studies

To elucidate the therapeutic effect of GP-2250 in combination with Gemcitabine in vivo, tumor responses were analyzed in patient-derived xenograft (PDX) mouse models by comparing Gemcitabine monotherapy, GP-2250 monotherapy, their combination and vehicle-treated control. The reduction in tumor growth by GP-2250 as a single agent was not superior, as previously proven by Buchholz et al. (2017) [26] and confirmed in two PDX mouse models in the present study (Appendix A). Thus, GP-2250 monotherapy was only performed in the first four PDX models, and the results are displayed in the supplements. Both, Gemcitabine as a single agent and the combination of GP-2250 and Gemcitabine led to a significant decrease in tumor growth compared to controls in all analyzed PDX models (Figure 3). All controls showed progressive disease according to RECIST, reaching abortion criteria within 6 weeks in six of eight PDX models and consequently had to be terminated. Treatment with Gemcitabine monotherapy reduced tumor growth in comparison to the control group but was still classified as progressive disease according to RECIST in all PDX models, except Bo69, where a stable disease with a relative tumor volume of 1.07 ± 0.24 was achieved. However, the combination therapy of GP-2250 and Gemcitabine was significantly superior in reducing tumor growth in all eight PDX models compared with Gemcitabine alone. Indeed, in four of the eight PDX (Figure 3A,C,D,G), the combination treatment resulted in a partial response after 6–8 weeks (3 weeks in Bo80) with a relative reduction in tumor volume of 0.23 ± 0.06 in Bo 69, 0.46 ± 0.14 in Bo103, 0.45 ± 0.15 in Bo80 and 0.69 ± 0.07 in Bo85. Three of eight PDX models (Figure 3B,F,H) displayed stable disease after treatment with a combination of GP-2250 and Gemcitabine. The relative tumor volume ranged between 0.76 ± 0.06 in Bo73, 0.76 ± 0.10 in Bo70 and 1.17 ± 0.55 in Bo122. Only the PDX model Bo82 (Figure 3E) showed a progressive disease under combination treatment with a relative tumor volume of 2.61 ± 0.34.

### 3.3. Maintenance Therapy

The potential of combining GP-2250 plus Gemcitabine was further analyzed in a maintenance therapy setting in 8 PDX mouse models. To achieve initial tumor control, the animals were first-line treated with the combination of Gemcitabine and nab-Paclitaxel for 14 days. This period was followed by treatment with the combination of GP-2250 and Gemcitabine or Gemcitabine alone. The controls showed a progressive disease according to RECIST, reaching abortion criteria within 3 to 5 weeks. In all PDX models tested, the maintenance therapy with either Gemcitabine or a combination of GP 2250 and Gemcitabine led to a significant decrease in tumor growth compared to controls (Figure 4). However, maintenance therapy with Gemcitabine monotherapy showed progressive disease in six of eight PDX models after 3 to 6 weeks (Figure 4A–C,E,G,H). Only the PDX model Bo73 displayed stable disease with a relative tumor volume of 1.06 ± 0.34, while partial remission with a relative tumor volume of 0.63 ± 0.32 in the PDX model Bo85 was achieved (Figure 4D,F). In striking contrast, the combination therapy of GP-2250 and Gemcitabine resulted in partial response after 6 to 8 weeks in 4 of 8 PDX models with relative tumor volume of 0.18 ± 0.10 in Bo85, 0.29 ± 0.20 in Bo73, 0.39 ± 0.17 in Bo69, and 0.66 ± 0.43 in Bo84 (Figure 4D–G). Additionally, three of eight PDX models showed stable disease after treatment with the combination of GP-2250 with Gemcitabine. The relative tumor volume ranged between 0.72 ± 0.46 in Bo57, 0.82 ± 0.20 in Bo66, and 0.86 ± 0.15 in Bo81 (Figure 4A,C,H). Only the PDX model Bo6 showed progressive disease with a relative tumor volume of 2.18 ± 0.26. However, the relative tumor volume was still significantly reduced compared to Gemcitabine monotherapy (5.13 ± 3.33), which reached the abortion criteria after 4 weeks (Figure 4B). Additionally, no significant changes in body weight (BW) were observed even after long-term maintenance therapy (Appendix A–J). Table 3 shows an overview of all in vivo results according to RECIST criteria. 

In summary, in the first-line PDX study, Gemcitabine alone did not result in tumor regressions in any of the eight patient-derived xenografts, whereas Gemcitabine plus GP-2250 led to tumor regressions in seven (87.5%) of the eight models. The aggregate tumor regression response in individual xenografts derived from the eight parental cases was six (10%) of 58 tumors for Gemcitabine as a single agent and 55 (74%) of 68 tumors for Gemcitabine plus GP-2250 (Figure 5A).

In the maintenance treatment model, Gemcitabine alone resulted in tumor regressions in only one (12.5%) of eight patient-derived xenografts, while Gemcitabine plus GP 2250 resulted in tumor regression in seven (87.5%) of eight cases. The aggregate tumor regression response in individual xenografts derived from the eight parental cases were 14 (19%) of 74 for Gemcitabine and 60 (79%) of 76 and Gemcitabine plus GP-2250 (Figure 5B).

### 3.4. Immunhistochemival Validation of Therapy Response

For the validation of the therapy response immunhistochemical H and E staining of the treated tumors of the first-line therapy was performed and the regression grades were determined. The results in Figure 6 show clearly, that the tumors which received the combined treatment of GP-2250 and Gemcitabin reached higher regression grades (grade 2 and 3 Le Scodan, grade 2 and 1 CAP) compared to those who received the individual therapies (grade 1 and 2 Le Scodan, grade 3 and 2 CAP).

### 3.5. Effect of GP-2250 on CD133^+^ Cells in Spheroid Cultures

To obtain a first assessment of the effect of GP-2250 and Gemcitabine on CD133^+^ cells, spheroids of the cell lines PancTuI and Bo80 were generated and analyzed for their expression of CD133^+^ on their surface under treatment compared to untreated controls. Concentrations of GP-2250 (1000 µM) and Gemcitabine (30 mM) with high cytotoxic effects after 48 h were chosen. Cultures of PancTuI and Bo80 showed stable and compact multi-cellular spheroid formation after an incubation period of 5–9 days (Figure 7A,B). The subsequent flow cytrometric analysis of these cultures revealed different ratios of CD133^+^ cells between both cell lines. As expected, the established cell line PancTuI (6.8%) showed an approximately 10× smaller fraction of CD133^+^ cells in their spheroid cell populations compared to the primary cell line Bo80 (67.7%). After incubation with GP-2250 the ratio of CD133^+^ cells was significantly reduced by 76.1 ± 18.7% in spheroid cell populations of PancTuI, and by 56.3 ± 13.5% in spheroid cell populations of Bo80 (Figure 7C,D) compared to the untreated control. No significant changes in the level of CD133^+^ cells were observed in spheroid populations treated with Gemcitabine. These data revealed that only substance GP-2250 was able to reduce the ratio of CD133^+^ markers on the surface of the cells in both established and primary cell line populations significantly. Conversely, in Gemcitabine-treated cultures, the ratio of CD133^+^ cells did not diminish at all. Morphologically, it was observed that spheroid cultures treated with GP-2250 show signs of dissociation of the spheroid structure indicating a certain amount of cell damage. This effect was not observed in spheroid cultures treated with Gemcitabine alone, even at such a high concentration of 30 mM (Figure 7E–G).

## 4. Discussion

The present study is the first to evaluate the combinatorial effect of GP-2250 and Gemcitabine on PDAC. As the prognosis for this disease is abysmal and established chemotherapeutic protocols are characterized by highly unfavorable side effect profiles, exploration of novel therapeutic options is indispensable in order to improve quality of life and survival in patients. The combination of these agents showed a strong synergism and proved to be highly effective in vitro on three cell lines derived from PDAC patient tissue and one established pancreatic cancer cell line. The advantage of this drug combination over Gemcitabine treatment as a single agent in first-line was confirmed in vivo in eight PDX mouse models which constitute a translational intermediate between cell culture and clinical trials. Similarly, promising results were found for the drug combination in a maintenance therapeutic setting following first-line treatment with nab-Paclitaxel plus Gemcitabine, as shown in eight additional PDX models. Furthermore, the combined treatment was not accompanied by weight loss and severe side effects, enabling a treatment period up to 60 days in mice. Finally, we were able to show that GP-2250 but not Gemcitabine significantly reduced the ratio of aggressive tumor-initiating CD133^+^ cells in spheroid cultures of both an established and a primary pancreatic cell line.

Pancreatic ductal adenocarcinoma is a highly malignant disease characterized by poor prognosis. Even in the case of early detection and after curative resection, pancreatic cancer relapses in more than half of patients [49]. As its incidence continuously increases, PDAC is projected to be the second-most lethal cancer type by 2030 [50]. Current German guidelines express an open recommendation for nab-Paclitaxel plus Gemcitabine or FOLFIRINOX as first-line therapies for PDAC, depending on the performance status of the patient [8]. In particular, FOLFIRINOX treatment is poorly tolerated and is accompanied by severe side effects such as peripheral neuropathy and gastrointestinal and hematological adverse reactions [9,10,11]. Therefore, only patients in good general condition qualify for administration. For patients with a lower performance status, Gemcitabine as monotherapy or in combination with Erlotinib is used [9,10]. German guidelines allow the application of Gemcitabine combined with nab-Paclitaxel in reduced dosage in cases of lower performance status (ECOG 2) [8]. Gemcitabine is an established chemotherapeutic agent approved for locally advanced and metastasized PDAC [51]. However, its efficacy is compromised by multiple mechanisms causing the development of secondary resistance [52,53,54,55]. One of those factors is overexpression of CD133. Already in 2007, Hermann et al. defined that CD133^+^ PDAC cells were associated with tumorigeneses and Gemcitabine resistance [34]. Furthermore, clinical studies related CD133 as an efficient prognostic factor correlated with poorer clinical outcomes, manifesting via advanced TNM stage, lesser tumor differentiation and early lymph node metastasis. In addition, high CD133 expression resulted not only in an unfavorable outcome, but it also promoted tumor recurrence after Gemcitabine treatment [32,33]. Therefore, the elimination of CD133^+^ cells may be a purposeful approach to counteract Gemcitabine resistance and tumor recurrence in PDAC patients.

Over the last few years, research on alternative first-line or maintenance regimens has been focused especially on targeted therapies which are limited to a very small number of patients fulfilling a narrowly defined set of criteria, i.e., particular mutations [56]. Hence, none of the novel antineoplastic agents have shown superior efficacy or survival benefits in larger randomized trials for the overall patient collective [57,58,59,60,61,62]. Moreover, uninterrupted full-dose chemotherapy often leads to cumulative toxicity without additional efficacy [13,14]. Thus, current options for maintenance are limited to a small group of eligible patients owing to the specific requirements of either mutation profile or good performance status. This underlines the crucial need to establish new therapeutic strategies and regimens with better tolerability for broad patients, including more vulnerable patients.

### 4.1. Choice of GP-2250 Plus Gemcitabine

The selection of the therapeutic agents for this study was focused on agents applicable as maintenance therapy with a favorable side effect profile. Therefore, this study combined the standard therapeutic agent Gemcitabine with the novel Oxathiazinane derivate GP-2250. GP-2250 has been proven to possess antiproliferative and antineoplastic properties on pancreatic cancer in vitro and in vivo [26,28]. Recently, similar to our results, Sofia et al. could show that tumor progression was significantly reduced by GP-2250 in a variety of human cancer cell lines in a xenograft mouse model [63]. Additionally, the substance exhibited strong synergistic effects in combination with Gemcitabine in vitro and in vivo in pancreatic neuroendocrine carcinoma [29]. The combination of GP-2250 with Gemcitabine is well tolerated as there were no changes in body weight or vital functions observed in the test animals ([26,29], Appendix A–J).

The combination of GP-2250 and Gemcitabine offers a therapeutic regime with high effectiveness and a low side effect profile rendering the regimen a potential option for patients with poor performance status. As no development of secondary resistances was observed, its applicability as a maintenance strategy seems promising. While monotherapy with Gemcitabine in vitro showed no significant reduction in cell viability, the combination of GP-2250 with Gemcitabine reduced cell viability by up to 70%. These results were confirmed in vivo, where a partial remission according to RECIST in seven of eight PDX models was achieved with the combination treatment. In contrast, Gemcitabine monotherapy resulted in progressive disease in six of eight PDX models.

The aggregate tumor regression response derived from the eight parental cases was 10% for Gemcitabine, whereas the combination achieved a regression response rate of 74%. Comparing our results to data by Von Hoff et al., the effects of Gemcitabine monotherapy are similar (response rate 24% von Hoff versus 10% in our collective) [64]. In comparison to the current guideline-appropriate first-line therapy Gemcitabine and nab-Paclitaxel, GP-2250 and Gemcitabine proved to be more effective (tumor regression response rate of 55% for nab-Paclitaxel and Gemcitabine versus 74% for GP-2250 and Gemcitabine) [64,65]. The results of the immunhistochemical regression grade analysis of these tumors showing higher regression grades of tumors receiving the combination therapy also strengthens this hypothesis.

To improve efficacy and/or reduce side effects, several modifications in the nab-Paclitaxel and Gemcitabine regimen have been investigated. For instance, Wolfe et al. in 2021 demonstrated in heterotopic and orthotopic xenograft models that altering the schedule of Gemcitabine application prior to nab-Paclitaxel versus concurrent delivery delayed tumor growth without added toxicity [66]. Despite these improvements, progressive disease eventually occurred. Likewise, combining Gemcitabine with selective inhibitors of nuclear export (SINEs) such as Selinexor or KPT-330 significantly reduced tumor growth in pancreatic cancer xenografts; however, the disease continued to progress according to RECIST [67,68,69]. In another study, an additional combination of Selinexor with Gemcitabine and nab-Paclitaxel resulted in tumor regression in a PDX model of pancreatic cancer [70]. However, the study lacked the comparison with a control group treated with a combination of Gemcitabine and nab-Paclitaxel in the PDX model. In an orthotopic MiaPaca 2 model, Selinexor significantly enhanced the effect of Gemcitabine and nab-Paclitaxel; yet, all arms displayed progressive disease [70]. This contrasts with the results of our study achieving a partial response in seven of eight PDX models in first-line therapy (see Figure 5).

### 4.2. Maintenance Study

Overall, PDX studies in PDAC for maintenance strategies are rare. Clinical trials have featured two different maintenance strategies in advanced PDAC [15,16]. The first strategy is to reduce the number of chemotherapeutic agents when sustained tumor control is achieved [17,18,71]. The only prospective study is the phase II PRODIGE35-PANOPTIMOX by Dahan et al., comparing 5-FU and Leucovorin following 8 cycles of FOLFIRINOX vs. 12 cycles of FOLFIRINOX without changing the regimen [17]. Although the primary endpoint of 6 months PFS did not differ significantly between the two arms, the maintenance group had a longer median survival without deterioration in quality of life (11.4 vs. 7.2 months) [17]. Concomitantly, progression-free survival ranged from 6.8 to 8 months in other albeit retrospective maintenance studies pursuing a similar strategy [18,72,73].

The other maintenance approach consists of introducing another agent with a different mechanism of action, often mutation-targeted therapies [19,20,21,22,23,24]. Among these, the results of the POLO study form the basis for the guideline recommendation for Olaparib maintenance therapy in patients with BRCA germline mutations following platinum-based chemotherapy [8]. Golan et al. observed an increased PFS to 7.4 months under Olaparib therapy in eligible patients compared to 3.8 months in the control arm [19]. However, OS did not change significantly (18.6 vs. 18.1 months). Similarly, Reiss et al. observed a 12-month PFS rate of 54.8% in eligible patients treated with PARP inhibitor Rucaloprid and a median OS of 23.5 months [21]. As the study design did not include an untreated control, the data can only be compared to the literature.

Analogously to the clinical POLO trial, Roger et al. treated an orthotopic PDAC Mouse model with FOLFIRINOX followed by either Olaparib or a combination of PARP, ATR and DNA-PK inhibitors (PAD) compared with vehicle control [22]. Olaparib maintenance reduced tumor growth in comparison to the control, although the disease progressed in both groups (+20.7% vs. +41.1%). In contrast, treatment with PAD maintained stable disease (+18.1% tumor growth) and achieved complete remission of liver metastasis [22]. Notwithstanding these promising results, only 4.5–8% of PDAC patients in the Western world present with germline BRCA1/2 mutations and are eligible for treatment with PARP inhibitors [25].

### 4.3. CD133^+^ Cells

Although the combination of GP-2250 and Gemcitabine was superior to monotherapy in efficiency for PDAC in a PDX mouse model, the underlying mechanism of synergy remains to be determined. A possible explanation for this promising interaction may lie in the reduction in the highly aggressive subpopulation of CD133^+^ cells in PDAC by GP-2250. In this study, GP-2250 but not Gemcitabine was able to reduce the ratio of CD133^+^ cells in PDAC spheroids of an established and a primary cell line (Figure 7C,D). Recently, the mechanism of action of GP-2250 was discovered [28]. It targets the glucose metabolism by inhibition of Hexokinase 2 (HK2) and GAPDH, resulting in AMPK activation due to a reduction in ATP and NADH levels ultimately inducing apoptosis via the inhibition of NFκB and mTOR signaling pathways [28].

Aberrant metabolism and a high energy demand is a hallmark of aggressive cancer cells and an upregulation of the glucose metabolism is frequently observed in cancer cells resistant to chemotherapy [74,75,76]. Shukla et al. found an upregulation of glycolysis by increased MUC1 expression in Gemcitabine-resistant pancreatic cancer cells causing a stabilization of HIF-1α resulting in turn in a higher dependency on glucose and an increased dNTP production [76,77]. Moreover, they were able to increase the Gemcitabine sensitivity of resistant pancreatic cancer cells through glucose deprivation and inhibition of HK2 with 2-Deoxy-Glucose (2DG) [77]. Therefore, the inhibition of HK2 via GP-2250 might be one possible explanation for the synergy between Gemcitabine and GP-2250 and the absence of Gemcitabine resistance in combined therapy, even after long-term treatment.

More specifically, Nomura et al. found an increase in glucose uptake and increased glycolysis in the CD133^+^ cell population of KPC cells [35]. Additionally, the activity of HK2 was found to be threefold increased, along with overexpression of TCA-cycle enzymes. They were able to show, that the inhibition of glycolysis in CD133^+^ cells augmented their sensitivity towards chemotherapeutic substances such as Gemcitabine [35]. Furthermore, Banerjee et al. demonstrated a 2.5-fold increase of NFκB binding in CD133^+^ cells compared to CD133^-^ cells, ensuring survival, increasing invasion, and evading apoptosis [78]. Gemcitabine reduced only the viability of CD133^-^ cells, while the highly aggressive tumor initiating CD133^+^ subpopulation remained unaffected [36].

Thus, with its ability to reduce glucose metabolism and inhibit NFkB, GP-2250 might preferentially target CD133^+^ cells and/or reduce the amount of CD133 in the cells, thereby sensitizing the tumor in whole to Gemcitabine. In keeping with this view, there was no major sign of resistance development and tumor recurrence with the drug combination in the maintenance in vivo study. Gemcitabine alone maintained tumor regression in only one of eight patient-derived xenografts, while Gemcitabine plus GP-2250 sustained tumor regression in seven of eight cases. Similarly, the aggregate tumor regression response for maintenance in individual xenografts derived from the eight parental cases was 19% for Gemcitabine monotherapy and 79% for Gemcitabine plus GP-2250 (Figure 5B). Nevertheless, this experiment was a purely exploratory first assessment to characterize the effects on CD133 surface expression specifically based on the individual substances. Further studies will explore the impact of GP-2250 on CD133^+^ cells and its contribution to the synergy between GP-2250 and Gemcitabine in displaying an excellent sustained response in combination therapy.

### 4.4. Limitations of the Study

Our data present highly promising preclinical data for the therapy of PDAC and contributes to the further examination and characterization of GP-2250, but some limitations need to be addressed.

Firstly, while a patient-derived xenograft model is a valuable translational model, since cancer tissue directly from patients is used, it has to be kept in mind that the extrapolation of PDX results to humans is limited [40]. Especially as it is necessary to use immunocompromised mice in order to allow propagation of the tissue, no statements on any possible side effects concerning the immune system can be made. Here, the current open-label phase 1/2 clinical trial to evaluate the safety and tolerability of GP-2250 [31] will provide the relevant information. Furthermore, the examination of CD133 in this study was a purely exploratory first assessment to characterize the effects mainly based on the individual substances. From our data, it cannot clearly be stated if GP-2250 is actually selectively eliminating CD133^+^ cells of the tumor population or if it functions in a negative regulatory way on the expression of CD133 on the surface of the cells. So, the exact mechanism of this promising synergy will be a prospect of further extensive studies.

## 5. Conclusions

As the prognosis for PDAC is dire, the further exploration of novel therapeutic options is imperative. The combination of GP-2250 and Gemcitabine proved to be highly effective for the treatment of PDAC in a preclinical first-line strategy, as well as in this first maintenance setting. The present study represents a new highly effective treatment strategy with the combination of GP-2250 plus Gemcitabine and is a first proof of concept in a maintenance setting of this promising combination in PDAC. Accompanied by the currently running clinical trial in the USA [31] and the favorable side effect profile of GP-2250, this forms the translational basis for future clinical trials, particularly for maintenance therapy with this promising drug combination. By testing 13 different PDX-models in total we attempted to take the high inter- and intra-tumoral heterogeneity PDAC into account. The preliminary data on the impact of GP-2250 on CD133^+^ cells open a new avenue to assess the mechanism of synergy between GP-2250 and Gemcitabine.

## Figures and Tables

**Figure 1 cancers-16-02612-f001:**
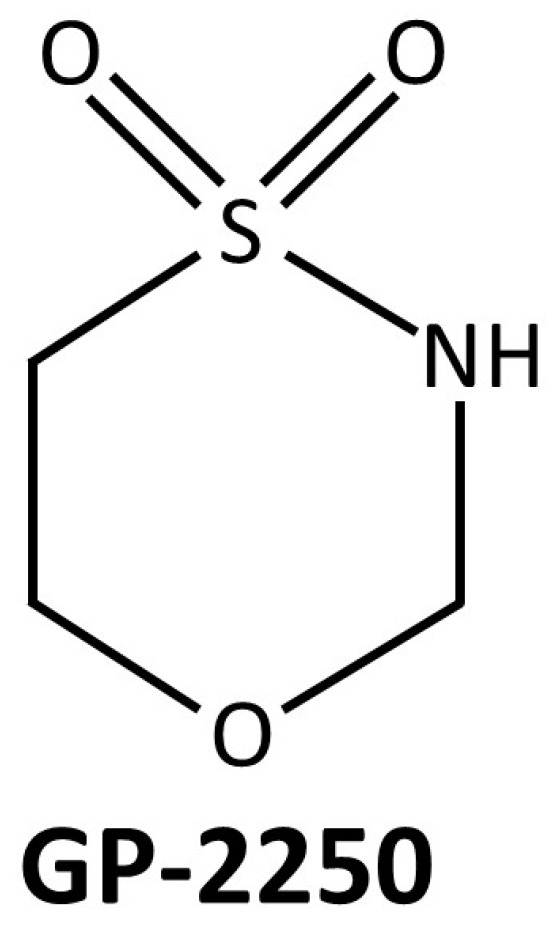
Molecular structure of the substance GP-2250 (Misetionamide), which is an Oxathiazinane derivative (1.4.5-oxathiazinan-dioxide-4.4). The metabolic half-life in blood of nude mice is 13.8 h [26].

**Figure 2 cancers-16-02612-f002:**
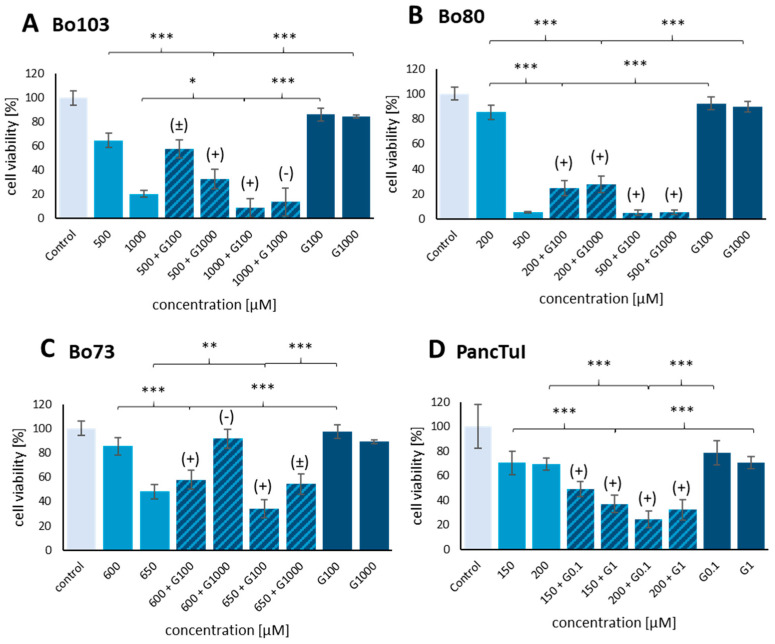
Cell viability by MTT analysis after 48 h of combination treatment. The graphs show the ratio of viable cells of (**A**): Bo103 treated with GP-2250 + Gemcitabine (G), (**B**): Bo80 treated with GP-2250 + Gemcitabine, (**C**): Bo73 treated with GP-2250 + Gemcitabine, (**D**): PancTuI treated with GP-2250 +Gemcitabine. Values are the mean ± SD of at least six independent experiments with consecutive passages. Concentrations are given in µM with G refers to Gemcitabine and GP refers to GP-2250. Asterisk symbols indicate the statistical significance of the decrease in viability compared with the untreated control (Control). *** *p* ≤ 0.001, ** *p* ≤ 0.01, * *p* ≤ 0.05. The combination indices (CI) by Chou–Talalay are marked as follows: CI < 0.9: synergistic effect (+), CI = 0.9–1.1: additive effect (±), and CI > 1.1: antagonistic effect (-).

**Figure 3 cancers-16-02612-f003:**
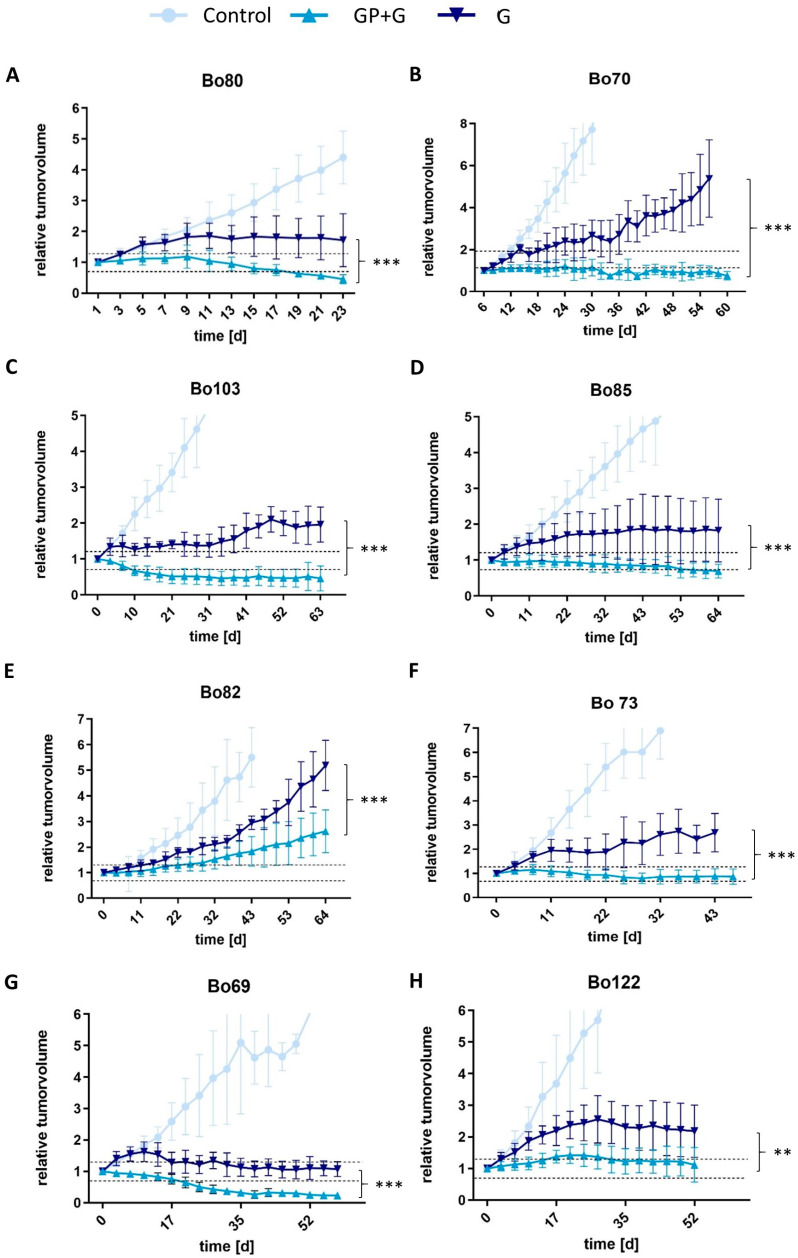
First-line therapy in vivo. Effects of 500 mg/kg*BW GP-2250 in combination with Gemcitabine (GP + G) (50 mg/kg*BW) versus Gemcitabine alone (G) versus vehicle-treated control on the subcutaneous tumor growth in nude mice in vivo. Nude mice with tumors of Bo80 (**A**), Bo70 (**B**), Bo103 (**C**), Bo85 (**D**), Bo82 (**E**), Bo73 (**F**), Bo69 (**G**), and Bo122 (**H**) were treated with GP-2250 (500 mg/kg*BW) three times a week in combination with Gemcitabine (50 mg/kg*BW) twice weekly, Gemcitabine monotherapy (50 mg/kg*BW) twice weekly or vehicle (control) for up to 9 weeks. The tumor volume was measured twice weekly. Asterisk symbols indicate differences between the combination treatment and the Gemcitabine monotherapy. *** *p* ≤ 0.001, ** *p* ≤ 0.01 (one-way ANOVA followed by Tukey’s post hoc test).

**Figure 4 cancers-16-02612-f004:**
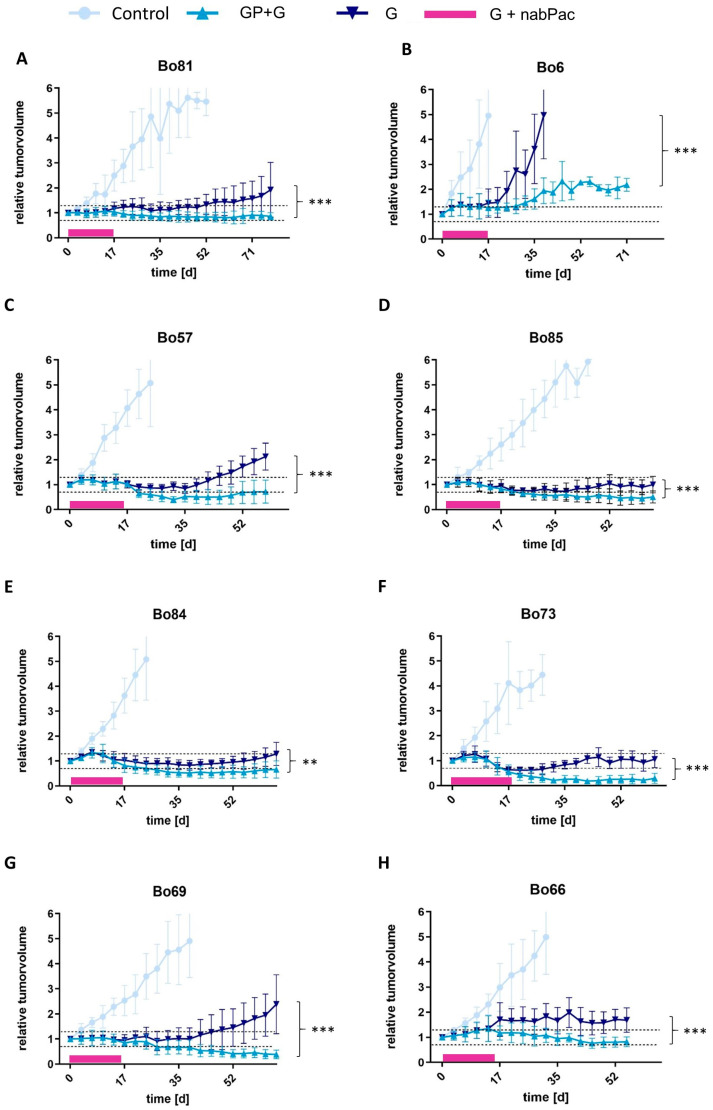
Maintenance therapy in vivo. Effects of the maintenance therapy with 500 mg/kg*BW GP-2250 in combination with Gemcitabine (50 mg/kg*BW) (GP + G) versus Gemcitabine alone (G) after treatment with Gemcitabine (100 mg/kg*BW) in combination with nab-Paclitaxel (30 mg/kg*BW) (G + nabPac) for two weeks on the subcutaneous tumor growth in nude mice in vivo. Nude mice with tumors of Bo81 (**A**), Bo6 (**B**), Bo57 (**C**), Bo85 (**D**), Bo84 (**E**), Bo73 (**F**), Bo69 (**G**), and Bo66 (**H**) were incubated after initial therapy with Gemcitabine monotherapy (50 mg/kg*BW) twice weekly, GP-2250 (500 mg/kg*BW) three times a week in combination with Gemcitabine (50 mg/kg*BW) twice weekly, or treated with vehicle (control) for up to 9 weeks. The tumor volume was measured twice weekly. Asterisk symbols indicate differences between the combination treatment and the Gemcitabine monotherapy. *** *p* ≤ 0.001, ** *p* ≤ 0.01 (one-way ANOVA followed by Tukey’s post hoc test).

**Figure 5 cancers-16-02612-f005:**
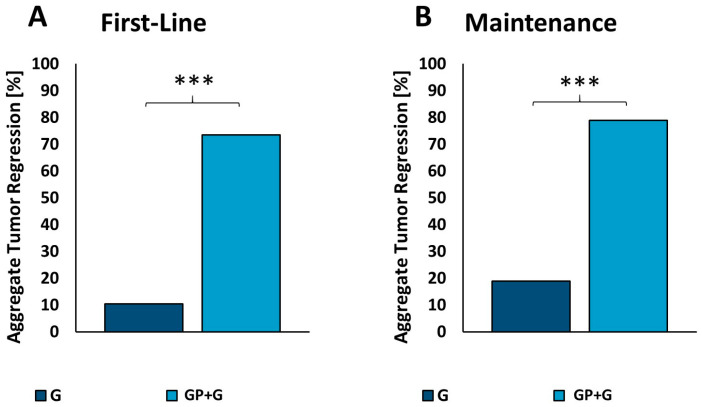
Aggregate tumor regression rate of the first-line therapy (**A**) and the maintenance therapy (**B**) model is shown in comparable bar graphs. *** *p* ≤ 0.001 (Fisher´s exact test).

**Figure 6 cancers-16-02612-f006:**
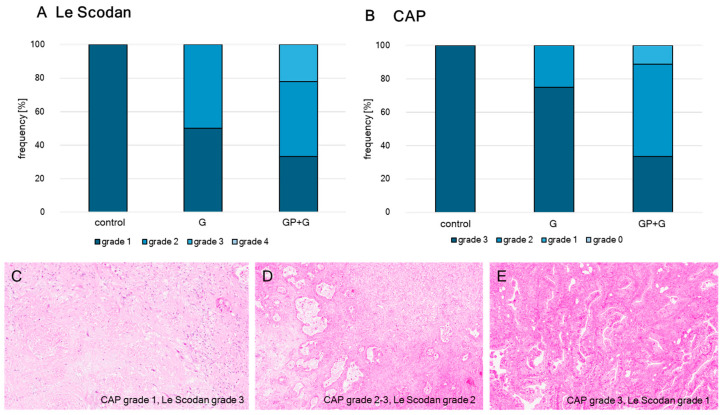
Distribution of tumor regression grades according to (**A**) Le Scodan and (**B**) CAP: Tumors of untreated controls (control) show lowest regression rates (Le Scodan 1, CAP 3, 10× (**C**)), where tumors treated with monotherapy Gemcitabine (G) show higher grades (grade 1 and 2 Le Scodan, grade 3 and 2 CAP, 5× (**D**)). Tumors treated with a combination of GP-2250 and Gemcitabine reached the highest regression grades (grade 2 and 3 Le Scodan, grade 2 and 1 CAP, 5× (**E**)).

**Figure 7 cancers-16-02612-f007:**
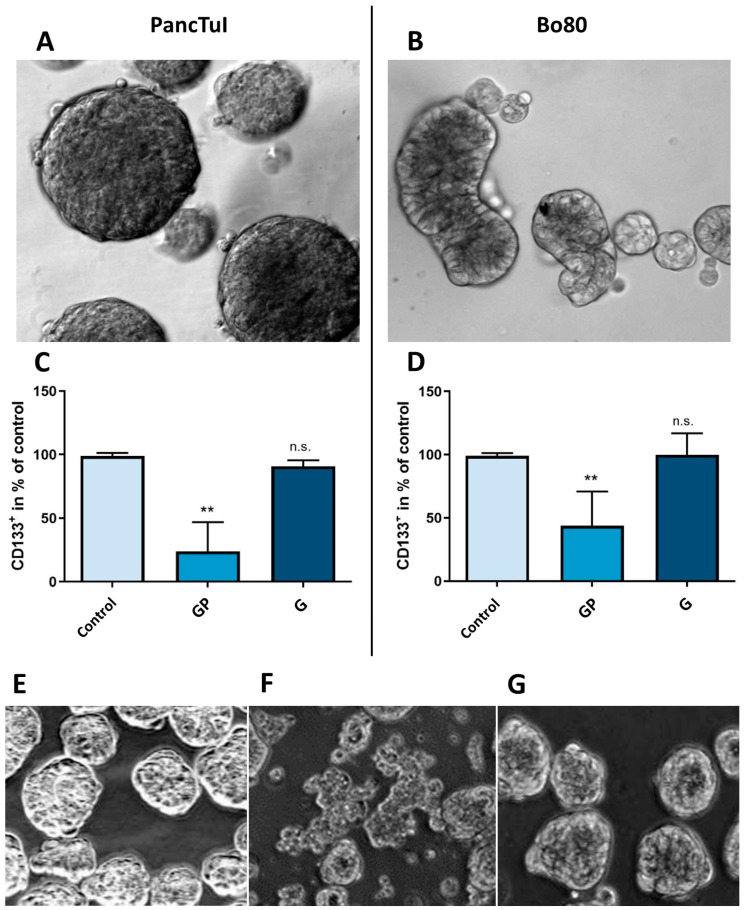
Effects of GP-2250 and Gemcitabine on CD133+ cell population in spheroid cultures of PancTuI and Bo80. Untreated spheroid cultures of PancTuI (**A**), Bo80 (**B**) showed different ratios of CD133^+^ cells in their population as shown by flow cytometry. Cells treated with Gemcitabine (30 mM) (G) or GP-2250 (1000 µM) (GP) for 48 h were analyzed via flow cytometry and compared to untreated control spheroid cultures which were taken as 100%. Shown is the normalized percentage of CD133^+^ cells PancTuI (**C**), and Bo80 (**D**) ±SD in %. Asterisk symbols indicate differences between the treated cultured and untreated control cells. ** *p* ≤ 0.01, n.s. *p* > 0.05 (one-way ANOVA followed by Tukey’s post hoc test). (**E**–**G**) Morphological effect of GP-2250 (100 µM) (**F**) and Gemcitabine (30 mM) (**G**) compared to an untreated control of Bo80 Spheroid cultures (**E**), all analyzed in 10× magnification.

**Table 1 cancers-16-02612-t001:** Composition of used media.

DMEM	DMEM
	10% FCS
	100 U/mL Penicillin
	100 µg/mL Streptomycin
	2 mM L-Glutamine
D-10 Medium	DMEM
	10% FCS
	200 U/mL Penicillin
	200 µg/mL Streptomycin
	2 mM L-Glutamine
	1 mM Sodium Pyruvate
DMEM/F12 modified	50% DMEM/F12
	50% D-10 Medium
	100 U/mL Penicillin
	100 µg/mL Streptomycin
	1.6 µg/mL Amphotericin
	10 µM Y27632 1HCl
	10 µg/mL Ciprofloxacin
	8.4 ng/mL Choleratoxin
	10 µg/mL Insulin
	20 nM 1-Thioglycerol
Spheroid Medium	DMEM/F12
	100 U/mL Penicillin/Streptomycin
	1× MEM NEAA
	1:50 B27 supplements
	100 µg/mL bFGF

**Table 2 cancers-16-02612-t002:** Classification and features of all analyzed tumors.

PDX ID	TNM Classification	Union Internationale Contre le Cancer (UICC) Stage
Bo69	pT3 pN1 M0 L1 V1 Pn1	UICC IIb
Bo70	pT3b pN1 M0 L0 V1 Pn1	UICC IIb
Bo73	pT3 pN1 M0 L1 Pn1	UICC IIb
Bo80	pT3 pN1 M0 L1 V1 Pn1	UICC IIb
Bo82	pT3 pN1 M0 L0 Pn1	UICC IIb
Bo85	pT3 pN0 M0 L1 V1 Pn1	UICC IIa
Bo103	pT3 pN1 M0 L1 V1 Pn1	UICC IIb
Bo122	pT3 pN2 M0 L1 V1 Pn1	UICC III
Bo6	pT2 pN1 M0 L1 V1 Pn1	UICC IIb
Bo81	pT3 pN2a M0 L1 V0 Pn1	UICC IIb
Bo84	pT3 pN1 M0 L1 V1 Pn1	UICC IIb
Bo66	pT3 pN0 M0 L1 V0 Pn1	UICC IIa
Bo57	pT3 pN0M0 L1 V0 Pn1	UICC IIa

**Table 3 cancers-16-02612-t003:** Results of the first-line and maintenance therapy in vivo. (PD: progressive disease, PR: partial response; SD: stable disease).

	First-Line			Maintenance	
Tumor	G	GP + G	Tumor	G	GP + G
Bo80	PD	PR	Bo81	PD	SD
Bo70	PD	SD	Bo6	PD	PD
Bo103	PD	PR	Bo57	PD	SD
Bo85	PD	PR	Bo85	PR	PR
Bo82	PD	PD	Bo84	PD	PR
Bo73	PD	SD	Bo73	SD	PR
Bo69	SD	PR	Bo69	PD	PR
Bo122	PD	SD	Bo66	PD	SD

## Data Availability

The data presented in this study are available on request from the corresponding author.

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
