# Peer review of "Maintenance Therapy for Pancreatic Cancer, a New Approach Based on the Synergy between the Novel Agent GP-2250 (Misetionamide) and Gemcitabine"

_cancers, 2024, doi:10.3390/cancers16142612_

Round 1

Reviewer 1 Report (Previous Reviewer 1)

Comments and Suggestions for Authors

The Authors have much improved the manuscript.

This manuscript is a resubmission of an earlier submission. The following is a list of the peer review reports and author responses from that submission.

Round 1

Reviewer 1 Report

Comments and Suggestions for Authors

In a study by Buchholz et al., the Authors investigate the activity of the the GP- 40 2250/Gemcitabine combination in maintenance therapy following nab-Paclitaxel plus Gemcitabine as first-line treatment in pancreatic cancer.

Specific comments:

1. The major obstacle is the use of very high concentration of GP-2250.

2. The study lacks any mechanistic investigation, it just reports potential drug combination based on simple in vitro assay and in vivo setting. In this respect, the study must be extended to support the conclusions. The observations are interesting, however, the study design should be extended.

3. Figure are hardly legible. 

4. Fig. 2. - CI values are not shown in the figure therefore, the last sentence from the figure legend should be transfer to Methods or to the description of the corresponding results.

5. Fig. 1 is unnecessary if the structure of this drug has been already published elsewhere.

Reviewer 2 Report

Comments and Suggestions for Authors

 Buchholz and the group have studied the combination of GP-2250 and gemcitabine's effect in cell lines and PDX models. They have achieved significant tumor suppression at lower doses of both agents. The data is excellent and can be published after major revisions. 

1. For combination experiments, calculate the synergy index using online software programs such as Synergy Finder. This will explain whether both agents are synergistic or additive. 

2. I suggest using a gemcitabine-resistant cell line and checking if GP-2250 can sensitize gemcitabine. 

3. The manuscript lacks mechanistic aspects. Please perform a few western blots or RTPCRs to identify the mechanism behind this more significant combination effect. 

4. I would check tumors for Ki67, Cl.caspase-3 etc to understand the antitumor effect. 

5. I suggest performing the H and E of tumors and vital organs. 

6. Is the GP-2250-GEM combination better than the GEM-nab paclitaxel?  

Comments on the Quality of English Language

i am not expert. 

Reviewer 3 Report

Comments and Suggestions for Authors

Specific comments to the authors

The authors Marie Buchholz et al. of the submitted manuscript "Maintenance therapy for pancreatic cancer, a new approach based on the synergy between the novel agent GP-2250 and Gemcitabine" investigated the synergistic effect of the novel oxathiazinane derivative GP-2250 and gemcitabine. To this end, the authors used the MTT cytotoxicity assay in vitro (using the pancreatic cell lines PancTuI and Bo73, Bo80 and Bo103 derived from PDAC patient-derived xenografts (PDX)) and analysis of tumour growth volume in vivo (using the PDX setting) after different drug treatment strategies.

In summary, the authors clearly demonstrated that the combination of GP-2250 and gemcitabine resulted in synergistic anti-tumour effects in vitro and in several in vivo pancreatic cancer models. Therefore, the authors postulated that the combination treatment strategy of GP-2250 and gemcitabine could be used as a possible maintenance therapy after nab-paclitaxel plus gemcitabine as first-line treatment for pancreatic cancer patients in the future.

Overall, the manuscript give interesting data on the novel Oxathiazinane derivative, GP-2250, in combination with the "standard" therapy gemcitabine in pancreatic cancer. The manuscript (including presentation) is comprehensible and convincing. The methods are mostly well described. Although the results and discussion are clearly presented, the authors (see specific comments) must perform some minor changes to improve the manuscript. In conclusion, the presented data are interesting. After incorporating the mentioned specific comments (see below) the manuscript has the potency to be accepted.

Specific comments

Material and Methods: Please explain the term “symbiotic effect” in relation to classical combinatory effects of synergistic and additive using isobolograms.

Results:

The numbering of the figure is wrong. Please correct.

# Figure 1: Please more data of pharmacodynamics and pharmacokinetics of the substance GP-2250 to the figure.

# Figure 2: The abbreviation of gemcitabine in figure 1D is not clear. Please explain in more detail.

# Figure 3 (“Figure 1: First-line therapy in vivo”): The differences between the controls, gemcitabine and the combination of gemcitabine and GP-2250 of all in vivo experiments could be presented in a summary manner, too.

# Figure 4 (“Figure 2: Maintenance therapy in vivo”): The differences between the controls, gemcitabine and the combination of gemcitabine and GP-2250 of all in vivo experiments could be presented in a summary manner, too.

# Figure 5 (“Figure 3: Aggregate tumor regression rate of the first-line therapy (A) and the maintenance therapy (B) model is shown in comparable bar graphs”): The significant differences are not indicated. Additionally to the aggregate data, a contingence table could display the recist data.

# Table 2: Please add information of grading to the table 2. What is the reason for selection of these PDXs. Please explain.

Discussion: The authors should discuss the limitations of the presented study, since definitive mechanistic insights of the combination of GP-2250 and gemcitabine are missing. Please discuss the findings in relation to following literature: Antineoplastic activity of GP-2250 in-vitro and in mouse xenograft models. Sofia RD, Martin KM, Costin JC. Anticancer Drugs. 2024 Feb 1;35(2):183-189. doi: 10.1097/CAD.0000000000001550. Epub 2023 Nov 21.  PMID: 37983375. How could the interesting findings transferred from a theoretical to a practical view (like clinical setting (drug-development, drug-combination))? Please discuss in short (in relation to the clinical trial “Trial to Evaluate Safety and Tolerability of GP-2250 in Combination With Gemcitabine”, NCT03854110, too).

Comments on the Quality of English Language

Minor editing of English language required.